# Mucosa-Associated Lymphoid Tissue 1 Is an Oncogene Inducing Cell Proliferation, Invasion, and Tumor Growth via the Upregulation of NF-κB Activity in Human Prostate Carcinoma Cells

**DOI:** 10.3390/biomedicines9030250

**Published:** 2021-03-03

**Authors:** Ke-Hung Tsui, Kang-Shuo Chang, Hsin-Ching Sung, Shu-Yuan Hsu, Yu-Hsiang Lin, Chen-Pang Hou, Pei-Shan Yang, Chien-Lun Chen, Tsui-Hsia Feng, Horng-Heng Juang

**Affiliations:** 1Department of Urology, Chang Gung Memorial Hospital-Linkou, Kwei-Shan, Tao-Yuan 33302, Taiwan; t2130@cgmh.org.tw (K.-H.T.); laserep@mail.cgu.edu.tw (Y.-H.L.); glucose1979@cgmh.org.tw (C.-P.H.); a9307@cgmh.org.tw (P.-S.Y.); clc2679@cgmh.org.tw (C.-L.C.); 2Department of Anatomy, College of Medicine, Chang Gung University, Kwei-Shan, Tao-Yuan 33302, Taiwan; D000016684@cgu.edu.tw (K.-S.C.); hcs@mail.cgu.edu.tw (H.-C.S.); hsusy@mail.cgu.edu.tw (S.-Y.H.); 3Graduate Institute of Biomedical Sciences, College of Medicine, Chang Gung University, Kwei-Shan, Tao-Yuan 33302, Taiwan; 4School of Nursing, College of Medicine, Chang Gung University, Kwei-Shan, Tao-Yuan 33302, Taiwan; thf@mail.cgu.edu.tw

**Keywords:** prostate, MALT1, NF-κB, CXCL5, IL-6, proliferation, tumor growth, invasion

## Abstract

Prostate cancer is one of the most common seen malignancies and the leading cause of cancer-related death among men. Given the importance of early diagnosis and treatment, it is worth to identify a potential novel therapeutic target for prostate cancer. Mucosa-associated lymphoid tissue 1 (MALT1) is a novel gene involved in nuclear factor κB (NF-κB) signal transduction by acting as an adaptor protein and paracaspase, with an essential role in inflammation and tumorigenesis in many cancers. This study investigated the functions and the potential regulatory mechanisms of MALT1 in the human prostate cancer cells. We found that MALT1 is abundant in prostate cancer tissues. MALT1 facilitated NF-κB subunits (p50 and p65) nuclear translocation to induce gene expression of interleukin 6 (IL-6) and C-X-C motif chemokine 5 (CXCL5) in prostate carcinoma cells. MALT1 promoted cell proliferation, invasion, and tumor growth in vitro and in vivo. MALT1 enhanced NF-κB activity in prostate carcinoma cells; moreover, NF-κB induced MALT1 expression determined by reporter and immunoblot assays, implying there is a positive feedback loop between MALT1 and NF-κB. In conclusion, MALT1 is a NF-κB-induced oncogene in the human prostate carcinoma cells.

## 1. Introduction

Prostate cancer (PCa) is the most frequently diagnosed malignancy and the sixth most common cause of cancer-related death in the world. In the U.S., PCa is the second leading cause of cancer death in American men [1]; while, in Taiwan, PCa is the sixth leading cause of cancer-related death according to the recent report from Department of Health, Taiwan [2]. Initially, PCa is dependent on androgens for growth. However, the androgen deprivation therapy provides only a transient control for recurrent prostate cancer that may result in progression from hormone-sensitive to aggressive castration-resistant PCa (CRPC) and androgen independence characterized by a resistance to such therapy [3]. Thus, it is needed to find the novel mechanisms related to the PCa progression, especially to those who are androgen-independent.

The nuclear factor κB (NF-κB) pathways is regarded as a link between inflammation and progression of certain types of cancer, including PCa [4,5,6]. It is well known that NF-κB activation is common in prostate carcinoma cells and is associated with the prostate malignancy [7]. Previous reports have indicated that sustained activation of NF-κB was found in androgen-independent prostate carcinoma cells [8,9,10]. Furthermore, one recent retrospective study using the immunohistochemical staining in 2 different multi-centre tissue microarrays indicated an association between the nuclear expression of NF-κB p65 and the PCa recurrence [11]. The studies concluded that NF-κB signaling pathways involved in the development of resistance against either castration or other androgen depletion treatments.

The mucosa-associated lymphoid tissue protein 1 (MALT1), a lymphoma oncogene functioning like a scaffold (adaptor protein) and paracaspase, regulates antigen receptor-mediated signal transduction in the NF-κB pathway [12,13,14,15,16,17]. Studies have focused on the MALT1 paracaspase activity and NF-κB signaling in the immunity of lymphoma, suggesting that dysregulation of this process may result in immune defects, autoimmune diseases, or cancers [18,19,20,21,22]. Therefore, therapeutic targeting of MALT1 protease activity is a potential approach for the treatment of lymphomas or as an effective strategy for treating those neoplastic and inflammatory disorders which associated with dysregulated NF-κB signaling [23]. Although the distinct mechanisms of MALT1 in cancer progression may be quite different from lymphomas, the oncogenic role and therapeutic potential of MALT1 have been well demonstrated in several non-lymphoid solid tumors, including breast cancer, lung cancer, melanoma, and cholangiocarcinoma [24,25,26,27]. However, the expression, biological function, and regulatory mechanisms of MALT1 in the human prostate have not yet been fully elucidated.

The aims of this study are to determine the expressions of MALT1 in both prostate carcinoma cells and prostate tissues, and to examine the potential functions and regulatory mechanisms of MALT1 in prostate carcinoma cells.

## 2. Materials and Methods

### 2.1. Cell Lines and Cell Culture

The cell lines of PZ-HPV-7, CA-HPV-10, LNCaP, PC-3, and DU145 were obtained from the Bioresource Collection and Research Center (BCRC, Hsinchu, Taiwan). PZ-HPV-7 and CA-HPV-10 cell lines are non-metastatic cells; while LNCaP, PC-3, and DU145 cell lines are metastatic prostate carcinoma cells as described previously [28]. The cells were maintained in RPMI 1640 medium (Life Technologies; Gaithersburg, MD) with 10% fetal calf serum (FCS; HyClone Laboratories, Inc. Logan, UT, USA) and incubated at 37 °C in a humidified 5% CO_2_ atmosphere. Caffeic acid phenethyl ester (CAPE) was purchased from Selleckchem (Houston, TX, USA). MI-2, a MALT1 inhibitor, was purchased from Tocris Bioscience (Bristol, UK). Phorbol myristate acetate (PMA), a NF-κB activator, was purchased from Sigma. Ionomycin, a membrane-permeable calcium ionophore, was purchased from InvivoGen (San Diego, CA, USA).

### 2.2. CyQUANT Cell Proliferation Assay

Cell proliferation was measured by the CyQUANT cell proliferation assay kit (Invitrogen, Carlsbad, CA, USA) as described previously [29].

### 2.3. EdU Flow Cytometry Assay

Cells were cultured in a serum-free medium for 24 h, then incubated for another 24 h in a 10% medium before incubation with EdU (5-ethynyl-2′-deoxyuridine; 10 µM) for 2 h. Subsequently, the cells were collected by trypsin-EDTA and centrifuged at 500 *g* for 10 min, then analyzed using Click-iT EdU Flow Cytometry Assay Kits (Thermo Fisher Scientific Inc. Waltham, MA, USA) as described previously [30].

### 2.4. Expression Vector Constructs and Stable Transfection

The expression vector containing the coding region of human *MALT1* cDNA (HG11618-UT) and control pCMV3 vector were purchased from Sino Biological Inc. (Beijing, China). The pCMV-IκBαM, an inhibitor of NF-κB expression vector, contains two mutations at residues 32 and 36 of serine to alanine that prevent phosphorylation (Cat. no. 631923; Clontech Laboratories, Inc.; Mountain View, CA, USA). The constructs of *NIK* (*MAP3K14*) expression vector (pcDNA-NIK) and control pcDNA3 expression vector were described in the previous studies [31]. The expression vectors were introduced into prostate carcinoma cells by electroporation using the ECM 830 (BTX, Holliston, MA, USA) set on a single pulse setting, 70 msec, and 180 V. The mock-transfection cells were transfected with control pCMV3 or pcDNA3 expression vectors and were clonally selected in the same manner as the gene-overexpression cells.

### 2.5. MALT1 CRISPR/dCas9 Lentiviral Activation Particles Transduction

Cloning the *MALT1*-overexpressed DU145 cells was performed using *MALT1* CRISPR/dCas9 lentiviral activation transduction particles, a synergistic activation mediator (SAM) transcription activation system for the transcriptional activation of endogenous genes, according to the manufacture’s protocol (Santa Cruz Biotechnology, Santa Cruz, CA, USA). Briefly, DU145 cells were plated onto 6-well plates 24 h before transfection. The culture media was replaced with RPMI-1640 medium plus 10% FCS and 5 µg/mL polybrene (Santa Cruz Biotechnology), then transduced with *MALT1* lentiviral activation transduction particles (sc-400791-LAC-2, Santa Cruz Biotechnology). Two days after transduction, the cells were selected by incubating with 10 µg/mL puromycin dihydrochloride for at least three generations. The mock-transfected cells were transduced with control lentiviral activation particles (sc-437282, Santa Cruz Biotechnology) and clonally selected in RPMI-1640 medium plus 10% FCS and 10 µg/mL puromycin dihydrochloride.

### 2.6. Gene Knockdown

Cells were plated onto 6-well plates 1 day before transfection. The culture media was replaced with RPMI-1640 medium plus 10% FCS and 5 µg/mL polybrene, then transduced with *MALT1* shRNA lentiviral transduction particles (sc-35845-V, Santa Cruz Biotechnology). Two days after transduction, the cells were selected by incubating with 10 µg/mL puromycin dihydrochloride for at least three generations. The mock-transfected cells were transduced with control shRNA lentiviral particles (sc-108080, Santa Cruz Biotechnology) and clonally selected in the same manner as the gene knockdown cells.

### 2.7. Nuclear and Cytoplasmic Extraction Assay

Nuclear and cytoplasmic fractions were separated using the NE-PERTM nuclear and cytoplasmic extraction kit (Thermo Fisher Scientific Inc.) as described previously [29].

### 2.8. Immunoblot Assays

Equal amounts of cell lysates were separated on 10% or 12% sodium dodecyl sulfate-polyacrylamide gels. The blotting membranes were probed using antiserum of GAPDH (6C5), MALT1 (EP603Y, Abcam, Cambridge, MA, USA), β-actin (MAB1501), NF-κB p50 (06-886), NF-κB p65 (06-418, Merck Millipore, Burlington, MA, USA), Lamin B1 (D9V6H), IκB-α (#9242), p-IκB-α (#2859, Cell Signaling Technology, Inc. Danvers, MA, USA), NDRG1 (42-6200, Thermo Fisher Scientific Inc.), or BTG2 [31]. Band intensities were detected by the Western lightning plus-ECL detection system (PerkinElmer Inc, Waltham, MD, USA), recorded using the LuminoGraph II (Atto Corporation, Tokyo, Japan), and analyzed using the GeneTools of ChemiGenius (Syngene, Cambridge, UK).

### 2.9. Real-Time Reverse Transcription-Polymerase Chain Reaction

The total RNA was isolated using Trizol reagent, and cDNA was synthesized using the superscript III (Invitrogen) system. Real-time polymerase chain reaction (qPCR) was performed using a CFX Connect Real-Time PCR system (Bio-Rad Laboratories, Inc., Singapore) as described previously [28]. FAM dye-labeled TaqMan MGB probes for *β-actin* (Hs01060665_g1), *BTG2* (Hs00198887_m1), *IL-6* (Hs00985639_m1), *CXCL5* (Hs01099660_g1), *MALT1* (Hs01120052_m1), and *NDRG1* (Hs00608387_m1) were purchased from Applied Biosystems (Foster City, CA, USA). Mean cycle threshold (Ct) values for target genes were normalized against *β-actin* to calculate delta Ct values.

### 2.10. Reporter Vector Constructs

The reporter vectors containing the 5′-flanking region of the *IL-6* genes were constructed as described previously [32]. The *NF-κB* reporter vector was purchased from Clontech Laboratories, Inc. The 5′-DNA fragment (−1 to −6313) of the human *MALT1* gene, according to the sequence from GenBank (AP005018.1), was synthesized by Invitrogen. The human *MALT1* reporter vector was constructed by cloning the DNA fragment into the pGL3-Basic reporter vector (pbGL3; Promega Biosciences, Madison, WI, USA) with *Hind III* sites. The DNA fragment containing the enhancer/promoter of the CXCL5 gene was isolated from the BAC clone (RP11-19B4, Thermo Fisher Scientific Inc.). A 9000 bp DNA fragment was subcloned into the pGEM-7 vector (Promega Biosciences) with *Pvu II*, then a DNA fragment containing the 5′ flanking region of the human *CXCL5* gene (–4782 to +23) was isolated by PCR with CXCL5 promoter primers (5′-ACAGCCCCATCC TTTTCTC-3′ and 5′-AGATCTACACTCATCTCTCC-3′). This fragment was then cloned into the luciferase reporter vector (pbGL3) at the *Bgl II* site.

### 2.11. Transient Transfection and Reporter Assay

The PC-3 cells were transiently transfected using the X-tremeGene HP DNA transfection reagent (Roche Diagnostics GmbH, Mannheim, Germany) as described previously [33]. The reactants were washed twice in PBS and the reaction was terminated by adding 200 µL of Luciferase Cell Culture Lysis Reagent (Promega Biosciences). The luciferase activity was determined in relative light units (RLU) using the synergy H1 microplate reader (BioTek Instruments, Inc., Beijing, China). Each sample was adjusted by the protein concentration of the whole-cell extract.

### 2.12. Enzyme-Linked Immunosorbent Assay

Cells were incubated with 0.5 mL RPMI medium in a 24-well plate for 24 h. After incubation, supernatants from each well were collected, and IL-6 and CXCL5 levels were analyzed using IL-6 enzyme-linked immunosorbent assay kit (#2107, Bioo Scientific Corporation, Austin, TX, USA) or CXCL5 enzyme-linked immunosorbent assay kit (DX000, R&D Systems, Inc., Minneapolis, MN, USA), respectively, as described by the manufacturer. Cell pellets were washed with ice-cold PBS, then dissolved in 200 µL PBS. After sonication, cell extracts were collected and protein concentrations were quantified by the BCA protein assay kit (Pierce, Rockford, IL, USA). The IL-6 and CXCL5 levels in the conditioned media were adjusted based on the protein concentration of the whole-cell extract.

### 2.13. Xenograft Animal Model

The studies were approved by the Chang Gung University Animal Research Committee (CGU106-157; 01/08/18) and performed per the United States National Institutes of Health Guide for the Care and Used of Laboratory Animals as described previously [28]. The four-week-old male nude mice (BALB/cAnN-Foxn1) were obtained from the animal center of the National Science Council (Taipei, Taiwan). PC3_shCOL and PC3_shMALT1 cells were detached from the cell flask by Gibco Versene solution (Life Technologies, Grand Island, NY, USA), washed with RPMI 1640 medium with 10% FCS, and were then re-suspended in PBS. The mice were anesthetized intraperitoneally, and 3 × 10^6^ cell/100 µL cells were injected subcutaneously on the wall of lateral back close to the shoulder of each mouse. The growth of tumors was measured by Vernier caliper every 2 to 3 days. The tumor volume was determined by the formula: Volume = Length × Width × Width/2. Mouse body weight was measured three times a week. After the mice were sacrificed, the tumors of xenograft animals were collected and digested with protein lysis buffer or Trizol reagent for further analysis of mRNA or protein expression of target genes.

### 2.14. Soft Agar Cloning Assay

The DU145-COL, DU145-MALT1-A-1, and DU145-MALT1-A-2 cells (50,000 cells each) were suspended in 1 mL of 0.4% agar (Difco Agar Noble, BD Biosciences) in RPMI1640 medium to prepare the top agar mixture. The mixture was plated on top of the base agar (0.8% in RPMI medium) in a 6-well plate for 4 weeks at 37 °C in a humidified 5% CO_2_ atmosphere. Cell colonies were observed by staining with 0.5 mL of p-iodonitrotetrazolium violet (1 mg/mL, Sigma) for 24 h, as described previously [34].

### 2.15. Matrigel Invasion Assay

The invasion ability of the cells was determined through an in vitro Matrigel invasion assay as described previously [28].

### 2.16. Immunohistochemical Assay

The human prostate tissue array was obtained from SuperBioChip Laboratories (Cat no: CA; Seoul, Korea). The pathologic staging and grading of tumors were performed based on the manufacturer’s datasheet. The tissue sections were stained with a human MALT1 primary antibody (anti-MALT1; 11660-1-AP, Proteintech Group, Inc. Rosemont, IL, USA) as descried previously [28]. Images were captured using a Paxcam 3 camera (PAX-it, Villa Park, IL, USA) and the intensity of the epithelia was scored (1 = weak; 2 = medium; 3 = intense) on the prostate lumen.

### 2.17. NF-κB (p65) Transcription Factor Binding Assay

The NF-κB (p65) binding activity was performed using NF-κB (p65) Transcription Factor Assay kit (Cayman Chemical, Ann Arbor, MI, USA). Briefly, the nuclear extracts were isolated as mentioned above, and incubated with consensus dsDNA sequence at 4 °C overnight. The samples were then incubated with p65 primary antibody for 1 h, followed by goat anti-rabbit HRP conjugate. The p65 binding activity was measured at absorbance 450 nm using the synergy H1 microplate reader (BioTek Instruments, Inc.) after treated with Transcription Factor Developing Solution.

### 2.18. Statistical Analysis

The results were expressed as the mean ± standard error (SE). The statistical significance was determined through paired t-test analysis and one-way ANOVA using the SigmaStat program for Windows version 2.03 (SPSS Inc., Chicago, IL, USA). Multiple comparisons were conducted using ANOVA with Tukey’s post hoc test. A significant difference was determined by *p*-value (*, # *p* < 0.05, **, ## *p* < 0.01).

## 3. Results

### 3.1. Expression of MALT1 in Prostate Carcinoma Cells and Prostate Tissues

To determine the correlation between MALT1 gene expression and metastasis in prostate cancer, the expression levels of MALT1 were characterized in five types of prostate cells. The metastatic prostate carcinoma cells (LNCaP, PC-3, and DU145) exhibited higher MALT1 protein (Figure 1A) and mRNA (Figure 1B) levels than non-metastatic cells (PZ-HPV-7 and CA-HPV-10), indicating that MALT1 expression in prostate carcinoma cells contributes to the malignancy in vitro. The results of RT-qPCR assays further indicated that MALT1 ubiquitously expressed in the prostate cancer cells. The immunostainings of MATL1 in normal (*n* = 9), grade II (*n* = 8), and grade III (*n* = 32) prostate cancer tissues were shown in Figure 1C. The quantitative intense scores of MALT1 immunostaining in the epithelium of the prostatic lumen in the normal prostate tissues were significantly lower than those in grade II and grade III prostate cancer tissues, with no significant differences between grade II and grade III prostate cancer tissues (Figure 1D), suggesting that MALT1 is a tumor marker for prostate cancer in vitro and in vivo.

### 3.2. Ectopic MALT1 Overexpression Induces IL-6 and CXCL5 via Activation of NF-κB Signaling to Enhance Cell Proliferation and Invasion In Vitro

Ectopic MALT1 overexpression in PC-3 cells induced not only MALT1 but also IL-6 and CXCL5 mRNA expressions as determined by immunoblot or RT-qPCR assays (Figure 2A). The reporter assays showed that transient overexpression of MALT1 enhanced the relative luciferase activities of both IL-6 and CXCL5 reporter vectors (Figure 2B). Furthermore, MI-2, a MALT1 inhibitor, or CAPE, an NF-κB inhibitor, attenuated the MALT1 induction of IL-6 and CXCL5 reporter activities (Figure 2C) and protein secretions (Figure 2D). The basic levels of IL-6 and CXCL5 in the medium were 40.78 ± 2.14 and 1451.52 ± 2.14 pg/µg cells, respectively. These results suggest that ectopic MALT1 overexpression enhances the expression of proinflammatory cytokines, such as IL-6 and CXCL5, via the NF-κB signaling pathway. The EdU flow cytometry assays revealed that ectopic overexpression of PC3-MALT1 cells increased cell proliferation by 6.5% compared to PC3-DNA cells (Figure 2E). Furthermore, Matrigel invasion assays showed that the invasion ability of PC3-MALT1 cells increased 2.4-fold compared to PC3-DNA cells (Figure 2F).

### 3.3. Upregulation of Endogenous MALT1 Expression Activates NF-κB to Enhance Tumorigenesis In Vitro

The induction of the transcriptional activation of endogenous MALT1 in DU145 cells by CRISPR/dCas9 lentiviral activation particles were used to confirm the biological functions of MALT1 in the prostate carcinoma cells. The immunoblot (Figure 3A, top) and RT-qPCR (Figure 3A, bottom) assays demonstrated that MALT1 was highly expressed in DU145_MALT1-Activation cells (DU145_MALT1-A1 and DU145_MALT1-A2) compared to mock-transduced DU145 cells (DU145_COL). Furthermore, either the anchorage-independent soft agar growth assays (Figure 3B, top) or the quantification of cell colonies (Figure 3B, bottom) revealed more tumorigenesis in DU145_MALT1-Activation cells in vitro compared to the mock-transduced control group (DU145_COL cells). Also, NF-κB signaling in DU145 cells was modified by MALT1. The immunoblot assays with nuclear and cytoplasmic extraction assays revealed that activation of endogenous MALT1 expression in DU145 (DU145_MALT1-Activation) cells upregulated phospho-IκBα protein levels, which destabilized IκBα protein levels in the cytoplasm and induced protein levels of p50 and p65 in the nucleus (Figure 3C). Taken together, these results suggest that activation of endogenous MALT1 facilitates NF-κB signaling, thereby enhancing tumorigenesis in vitro in DU145 cells.

### 3.4. Knockdown of MALT1 Attenuates Cell Proliferation and Cell Invasion In Vitro and Decreases Tumor Growth In Vivo

To further evaluate the role of MALT1 in the growth of prostate carcinoma cells, MALT1 was knocked down in PC-3 cells. The expressions of MALT1, IL-6, or CXCL5 in the selected clones determined by immunoblot (Figure 4A) or RT-qPCR assays (Figure 4B) indicating that MALT1 enhanced the expressions of IL-6 and CXCL5 in PC-3 cells. The CyQUANT cell proliferation assays revealed that MALT1-knockdown PC-3 (PC3_shMALT1) cells possessed much lower proliferative rates compared to mock-knockdown PC-3 (PC3_shCOL) cells during the 5-day incubation period (Figure 4C). Similar results were observed in the EdU proliferation assays (6% decrease) (Figure 4D) indicating that MALT1 knockdown attenuated cell proliferation. Furthermore, the Matrigel invasion assay showed that knockdown of MALT1 significantly inhibited cell invasion in PC-3 cells (Figure 4E). To evaluate the tumor growth effect of MALT1 in vivo, PC3_shCOL and PC3_shMALT1 cells were subcutaneously injected into the back and close to the shoulder of male nude mice (BALB/cAnN-Foxn1). The body weight and tumor volume were measured twice a week during the experimental period. Mice were sacrificed and tumors were collected on day 37 after inoculation. The results showed that tumors in the PC3_shCOL group grew rapidly in comparison to the PC3_shMALT1 group (201.11 ± 52.34 vs. 5.38 ± 1.31 mm^3^) during 37 days of experimental period (Figure 4F); however, the average body weight of the PC3_shMALT1 group was not significantly different to the vehicle-treated group (Figure 4G). The average tumor weight was reduced in the PC3_shMALT1 group compared to the PC3_shCOL group (0.0125 ± 0.0053 vs. 0.2342 ± 0.0669 g; Figure 4H). Further immunoblot assays confirmed that MALT1 was knocked down in the xenograft tumors derived from PC3_shMALT1 cells (Figure 4I). Meanwhile, consistent with the in vitro study, the mRNA levels of IL-6 and CXCL5 were downregulated in the xenograft tumors derived from PC3_shMALT1 cells compared to those derived from PC3_shCOL cells (Figure 4J). Collectively, these results suggest that knockdown of MALT1 in highly metastatic PC-3 cells significantly inhibits tumorigenesis in the xenograft mice model.

### 3.5. MALT1 Modulates NF-κB Activation in PC-3 Cells

The effect of MALT1 knockdown on the modulation of NF-κB signaling in PC3 cells was investigated. Nuclear and cytoplasmic fractions were separated and determined by Lamin B1 and GAPDH, respectively, showing that MALT1 protein was located predominantly in the cytoplasm (Figure 5A), with downregulation of p-IκBα protein expression in the cytoplasmic fraction and NF-κB (p50, p65) in the nuclear fraction, respectively. The quantitative analysis was illustrated in Figure 5B. Furthermore, the NF-κB activity was upregulated in the PMA/Ionomycin-treated PC-3 cells and blocked when MALT1 was knocked down in PC-3 (PC_shMALT1) cells compared with mock knockdown PC-3 (PC_shCOL) cells determined by the NF-κB (p65) transcription factor binding assay (Figure 5C). The reporter assays revealed that NF-κB promoter activity was increased in cells cotransfected transiently with the MALT1 expression vector (Figure 5D), confirming that MALT1 is the inductor of NF-κB signaling.

### 3.6. NF-κB Activity Modulates MALT1 Expression in PC-3 Cells

The MALT1 reporter vector cotransfected with NF-κB inducing kinase (NIK) expression vector upregulated reporter activity of MALT1; whereas, an inhibitor of the NF-κB expression vector (pCMV-IκBαM) blocked the reporter activity (Figure 6A). To prove that signal transduction of the NF-κB pathway was blocked by IκBα, we stably transfected the pCMV-IκBαM expression vector into PC-3 cells (PC3-IκBαM), confirming that IκBα but not phospho-IκBα was upregulated in the cytoplasmic fraction of PC3-IκBαM cells in comparison to the mock-transfected PC-3 (PC-DNA) cells; however, the protein levels of p65 and p50 in the nuclear fraction were downregulated by IκBαM overexpression (Figure 6B). Immunoblot (Figure 6C) and RT-qPCR (Figure 6D) assays revealed that MALT1 expression was downregulated by overexpression of IκBαM, but expressions of NDRG1 and BTG2 were upregulated in PC-3 cells. Taken together, these results demonstrate that a positive regulative loop exists between MALT1 and NF-κB activity.

## 4. Discussion

The NF-κB activation plays a key role in the development of AR antagonist resistance or CRPC [9]. The combination of NF-κB inhibition with AR inhibitor could be a promising target to prevent the evolution of prostate cancer [35,36]. MALT1 is regarded as a paracaspase with arginine-specific proteolytic activity similar to the caspases that enhancing NF-κB activation by cleavage of NF-κB negative regulators, such as TNFAIP3 (A20), RelB, and CYLD [37,38,39]. Studies suggested that MALT1 overexpression in lymphoma could result from gene translocation, form MALT1-fusion proteins, and activate the NF-κB pathway [20,21]. Recent studies found that MALT1 is required to activate the NF-κB signaling and promote cancer progression in the non-lymphoid system, such as breast cancer, lung cancer, melanoma, and cholangiocarcinoma [24,25,26,27].

A previous study concluded that cholangiocarcinoma patients with higher MALT1 expression exhibited poorer survival rates compared to those with lower MALT1 expression [27]. Furthermore, MALT1 gene silencing inhibits cell growth of melanoma in vitro and in vivo [26]. The RT-qPCR assays in this study showed that MALT1 exists in either metastatic or non-metastatic prostate cancer cells. The present study showed that metastatic prostate carcinoma cells (LNCaP, PC-3, and DU145) have higher MALT1 expression compared to non-metastatic cells (PZ-HPV-7 and CA-HPV-10), and MALT1 is abundant in advanced prostate cancerous tissues, suggesting that MALT1 could be an oncogene in the human prostate.

Previous studies verified that MALT1 is an inductor of NF-κB signaling pathway to promote cancer progression [21,22,24,25,26,27]. As shown in Figure 2 and Figure 4, overexpression or knockdown of MALT1 in prostate cancer PC-3 cells affected the oncogenic role of MALT1 in cell proliferation, migration, and invasion via the NF-κB pathway in vitro. The knockdown of MALT1 in highly metastatic PC-3 cells decreased cell growth in vitro and tumor growth in vivo. Furthermore, ectopic overexpression of MALT1 in DU145 cells not only induced NF-κB but also enhanced tumorigenesis in vitro. Taken together, these results indicate that MALT1 is an oncogene in prostate cancer in vitro and in vivo.

Previous studies indicated that IL-6 and CXCL5 could induce androgen-independent prostate cancer progression [40,41]. IL-6, a major cytokine in the tumor microenvironment, is highly expressed in almost all types of tumors and promotes tumorigenesis by regulating all hallmarks of cancers and multiple signaling pathways [42]. Previous studies indicated that MALT1 was required for the EGFR- and TNF-α/NF-κB signaling-induced IL-6 production in lung cancer progression [25,43]. Our study indicated that MALT1 induces both IL-6 and CXCL5 expressions in PC-3 cells, while knockdown of MALT1 decreases IL-6 and CXCL5 mRNA levels in PC-3 cells in vitro and in vivo (Figure 2 and Figure 4). The MI-2 (2-Chloro-N-[4-[5-(3,4-dichlorophenyl)-3-(2-methoxyethoxy)-1H-1,2,4-triazol-1-yl]phenylacetamide]) has been demonstrated as a MALT1 inhibitor which decreases MALT1-induced NF-κB activity in lymphoma (ABC-DLBCL) and cholangiocarcinoma cells [27,44]. The CAPE is well-known as the inhibitor of activation of NF-κB in human cancer cells [45]. The activation of MALT1 on the expressions and secretions of IL-6 and CXCL5 was blocked under the treatment of MI-2 or CAPE (Figure 2C,D), suggesting that upregulation of MALT1 on the IL-6 and CXCL5 expressions may rely on the MALT1/NF-κB pathway in prostate carcinoma cells. Collectively, our results illustrate that MALT1 modulates the expressions of IL-6 and CXCL5 via NF-κB activation, thereby contributing to tumor progression and malignancy in prostate carcinoma cells. However, study has shown that MALT1 modulated the IL6 mRNA stability [46]; therefore, the precise mechanism of MALT1 on IL-6 or CXCL5 gene expression still needs to be verified.

The NF-κB signaling is primarily regulated by the inhibitor κB (IκB) proteins and IκB kinase complex through the canonical and non-canonical NFκB pathways. The p50 and p65 NF-κB heterodimers are sequestered in the cytoplasm in an inactive state by IκBαM [47]. Our results showed that activation of endogenous MALT1 expression in DU145 cells upregulated phospho-IκBα protein levels, which destabilized IκBα protein levels in the cytoplasm and induced p50 and p65 protein levels in the nucleus. While, MALT1 knockdown in PC-3 cells demonstrated the opposite effect. Taken together, MALT1 induced p65 translocation into the nucleus suggesting that MALT1 may be implied as an important tumor marker for prostate since studies confirmed that nuclear localization of NF-κB p65 as a prognostic biomarker in prostate cancer patients [11,12,48]. The results showed that the in vitro tumorigenesis is higher in the DU145_MALT1-A1 cells than the DU145_MALT1-A2 cells since DU145_MALT1-A1 cells expressed more MALT1 and p-IκBα proteins, indicating that MALT1 levels is compatible with the capability of tumorigenesis in vitro. As shown in Figure 5, our study proved that PMA/Ionomycin treatment induced NF-κB (p65) activity in PC-3 cells, which was in agreed to previous reports [49,50]. Furthermore, knockdown of MALT1 downregulated NF-κB activity in PC-3 cells, confirming the modulation of MALT1 on NF-κB signaling in prostate carcinoma cells.

The NF-κB inducing kinase (NIK), also called mitogen-activated protein kinase kinase kinase 14, is a central regulator of non-canonical NF-κB signaling or alternative activation of canonical NF-κB signaling in response to the stimulation of TNF receptor superfamily members [5,51]. Previous report showed that overexpressed MALT1 in the MALT lymhpoma was capable of activating both canonical and non-canonical NF-κB pathways [16]. Our study confirmed that ectopic overexpression of NIK induces MALT1 gene expression. The results are in agreed with our previous studies which verified that ectopic overexpression of NIK via the NF-κB pathways upregulated migration and invasion enhancer 1 (MENI) but downregulated BTG2 in the prostate carcinoma cells [28,31]. Results of the present study also revealed that ectopic overexpression of IκBα downregulates the protein levels of p65 and p50 in the nuclear fraction and attenuates MALT1 gene expression. As shown in Figure 6, ectopic overexpression of IκBα induced NDRG1 and BTG2 expression is consistent with our previous studies [28,31]. The present study demonstrated that MALT1 is one of the key proteins modulating NF-κB signaling in prostate carcinoma cells. Furthermore, the evidence of present study suggesting a positive regulative loop exists between MALT1 and NF-κB activity in PC-3 cells.

## 5. Conclusions

In this study, we demonstrate that MALT1 facilitates nuclear translocation of NF-κB subunits, p50 and p65, to induce gene expression of IL-6 and CXCL5. The presence of novel positive feedback loop between MALT1 and NF-κB in prostate carcinoma cells indicates that MALT1 is a NF-κB-induced oncogene and the induction of NF-κB activation by MALT1 enhances cell proliferation, invasion, and tumor growth in the human prostate carcinoma cells in vitro and in vivo.

## Figures and Tables

**Figure 1 biomedicines-09-00250-f001:**
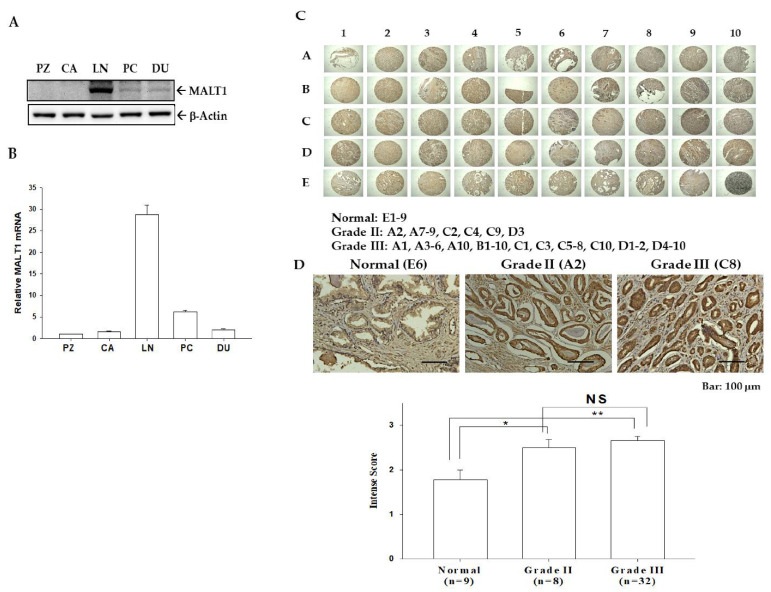
Expression of MALT1 in prostate carcinoma cells and prostate tissues. MALT1 expression in human prostate cells (PZ: PZ-HPV-10, CA: CA-HPV-7, LN: LNCaP, PC: PC-3, DU: DU145) was determined by (**A**) immunoblot and (**B**) RT-qPCR assays. (**C**) MALT1 expression levels were determined by IHC staining in human prostate tissue array with normal and cancer tissues. (**D**) The enlarged figures of normal, grand II, and grade III were shown in the top panel, and the intense score of MALT1 in normal (*n* = 9) or cancer tissues (grade II, *n* = 8; grade III, *n* = 32) were determined by IHC staining. * *p* < 0.05, ** *p* < 0.01, NS represents no significance.

**Figure 2 biomedicines-09-00250-f002:**
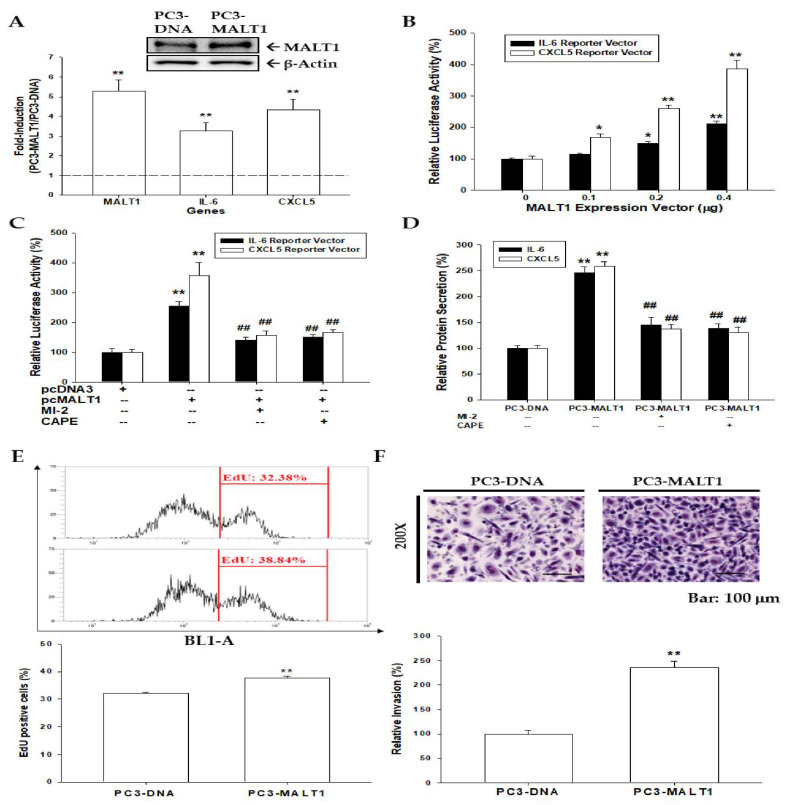
Effects of ectopic MALT1 overexpression on cell proliferation, cell invasion, and gene expression of IL-6 and CXCL5 in PC-3 cells. (**A**) Relative expressions of MALT1, IL-6, and CXCL5 after ectopic MALT1 overexpression in PC-3 cells were detected by immunoblot or RT-qPCR assays. (**B**) Relative luciferase activities of IL-6 and CXCL5 reporter vectors after cotransfection with various doses of MALT1 expression vectors. (**C**) The relative luciferase activities of IL-6 and CXCL5 reporter vectors after transient overexpression of MALT1 expression vector and treated with/without MI-2 and CAPE. Data are presented as mean percentage ± SE (*n* = 6) of the luciferase activity in relation to the vehicle-treated group (* *p* < 0.05, ** *p* < 0.01), or relative to that of the MI-2 and CAPE-treated group (## *p* < 0.01). (**D**) Secretion of IL-6 and CXCL5 in MALT1-overexpressed PC-3 cells treated with/without MALT1 inhibitor, MI-2, a NF-κB inhibitor, CAPE. (**E**) The cell proliferation rate of PC3-DNA and PC3-MALT1 cells was measured by the EdU flow cytometry assay (± SE, *n* = 4; ** *p* < 0.01). (**F**) The effect of MALT1 overexpression on cell invasion in PC-3 cells was determined by the Matrigel invasion assay (bar: 100 µm), (± SE, *n* = 3; ** *p* < 0.01).

**Figure 3 biomedicines-09-00250-f003:**
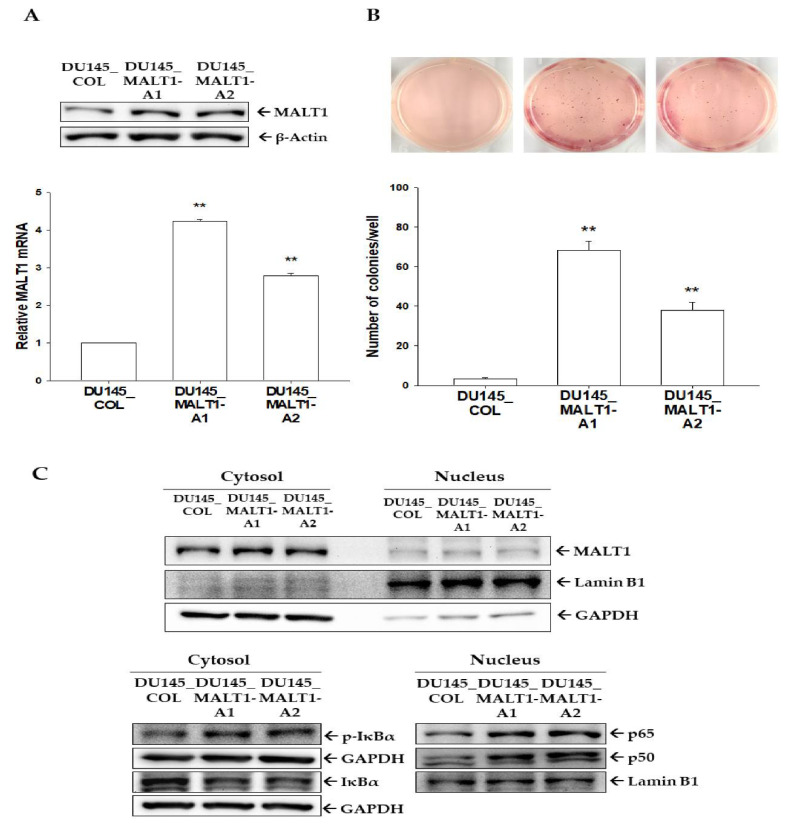
Modulation effect of MALT1 on tumorigenesis in vitro and NF-κB activity in prostate carcinoma DU145 cells. (**A**) Protein levels (top) and mRNA levels (bottom) of MALT1 after ectopic overexpression in DU145 (DU145_MALT1-A1 and DU145_MALT1-A2) cells. Data are presented as mean fold-induction of the mRNA levels (± SE, *n* = 3) relative to the mock-transduced group. (**B**) In vitro tumorigenesis of MALT1-overexpressed DU145 cells was determined by soft agar assays. The numbers of colonies were counted (bottom) under a microscope (± SE, *n* = 6). (**C**) Expressions of MALT1, IκBα, phospho- IκBα, p65, p50, Lamin B1, and GAPDH in cells after separation of nuclear and cytoplasmic fractions as indicated were determined by immunoblot assays. ** *p* < 0.01.

**Figure 4 biomedicines-09-00250-f004:**
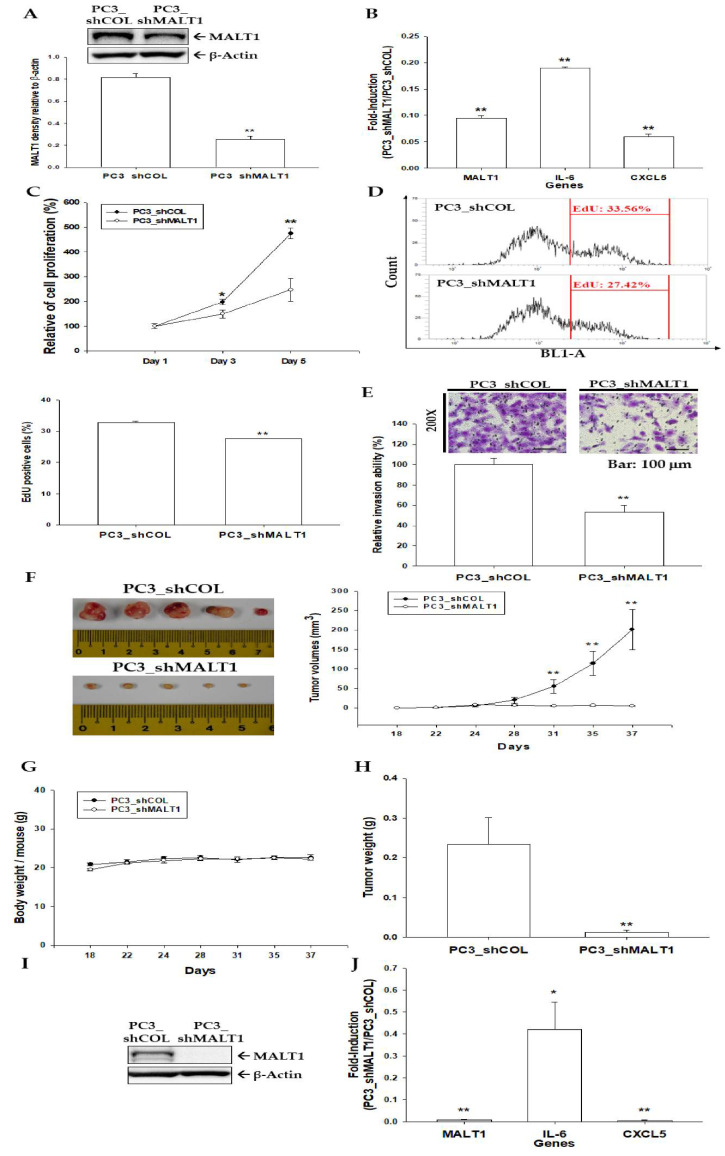
Effect of MALT1 knockdown on cell proliferation, invasion, and tumor growth in PC-3 cells in vitro and in vivo. (**A**) Relative expression of MALT1 in MALT1 knockdown PC-3 cells was assessed by immunoblot assays (top) and the data of relative density (*n* = 3) are expressed as the intensity of the MALT1/β-Actin (bottom). (**B**) Relative expressions of MALT1, IL-6, and CXCL5 in MALT1 knockdown PC-3 cells were assessed by RT-qPCR assays. Data are presented as mean fold-induction of the mRNA levels (±SE, *n* = 3) relative to the mock knockdown (PC3_shCOL) group. (**C**) Cell proliferation of MALT1-knockdown PC-3 (PC3_shMALT1) cells relative to the mock knockdown (PC3-shCOL) group was determined using the CyQUANT cell proliferation assays (±SE, *n* = 8). (**D**) The cell proliferation rate of PC3_shCOL and PC3_shMALT1 cells was measured by the EdU flow cytometry assays (±SE, *n* = 4). (**E**) Effect of MALT1 knockdown on cell invasion in PC-3 cells (bar: 100 µm), (±SE, *n* = 3). The athymic male nude mice were randomly divided into two groups. PC3_shCOL cells and PC3_shMALT1 cells (3 × 106) were injected subcutaneously into the dorsal area of the mice. The (**F**) tumor volumes (mm^3^ ± SE, *n* = 5) derived from PC3_shCOL cells (●, *n* = 5) and MALT1-knockdown (PC3_shMALT1) cells (○, *n* = 5), respectively, and (**G**) body weight (±SE, *n* = 5) of mice were measured every 2–3 days during 37 days. (**H**) The tumor weight (±SE, *n* = 5) was measured after sacrificed. (**I**) Protein levels of MALT1 in the tumors derived from PC3_shCOL and PC3_shMALT1 cells were determined using immunoblot assays. (**J**) Relative mRNA expressions of MALT1, IL-6, and CXCL5 in xenograft tumors were determined using RT-qPCR assays. Data are presented as mean fold-induction of the mRNA levels (±SE, *n* = 3) relative to the PC3_shCOL group. * *p* < 0.05, ** *p* < 0.01.

**Figure 5 biomedicines-09-00250-f005:**
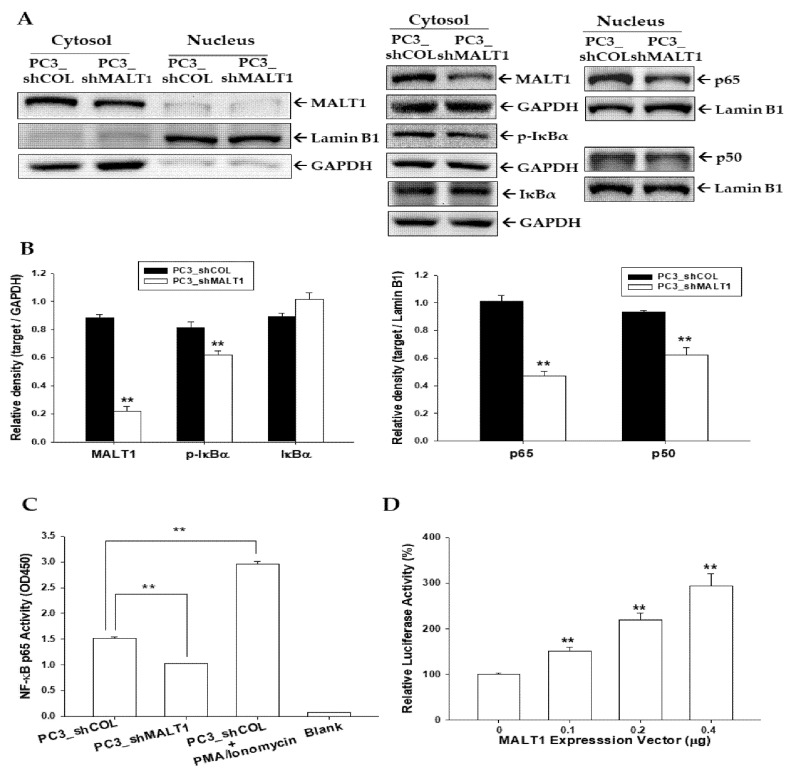
Modulation of MALT1 on NF-κB activity in prostate carcinoma PC-3 cells. (**A**) Expressions of MALT1, IκBα, phospho- IκBα, p65, p50, Lamin B1, and GAPDH in PC3_shCOL and PC3_shMALT1 cells after separation of nuclear and cytoplasmic fractions determined by immunoblot assays. (**B**) The data of relative density are expressed as the intensity of the protein band produced from the target gene/GAPDH or Lamin B1 as indicated (*n* = 3). (**C**) The NF-κB (p65) binding activity in PC3_shCOL cells, PC3_shMALT1 cells, and PMA/Ionomycin-treated PC-3 cells. ** *p* < 0.01. (**D**) The reporter activity of NF-κB reporter vector cotransfected with various dosages of the MALT1 expression vectors, as indicated in the PC-3 cells. Data are expressed as the mean percentage ± SE (*n* = 6) of luciferase activity relative to the mock-transfected group.

**Figure 6 biomedicines-09-00250-f006:**
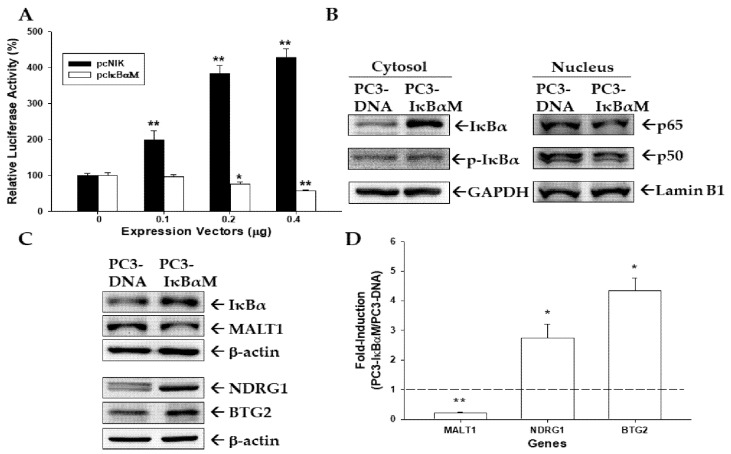
NF-κB activation upregulated MALT1 expression in prostate carcinoma cells. (**A**) The reporter activity of MALT1 reporter vector cotransfected with various dosages of NIK (black bars) or IκBαM (white bars) expression vectors as indicated in the PC-3 cells. Data are expressed as the mean percentage ± SE (*n* = 6) of luciferase activity relative to the mock-transfected group. (**B**) The mock-transfected PC-3 (PC3-DNA) and IκBαM-overexpressed PC-3 (PC3- IκBαM) cells were lysed, then IκBα, phospho-IκBα, p65, p50, GAPDH, or Lamin B1 after separation of cytoplasmic and nuclear fractions as indicated were determined by immunoblotting. The PC3-DNA and PC3-IκBαM cells were lysed, then IκBα, MALT1, NDRG1, and BTG2 were determined by immunoblot (**C**) and RT-qPCR (**D**) assays. Data are presented as mean fold-induction of the mRNA levels (± SE, *n* = 3) relative to the vehicle-treated group. * *p* < 0.05, ** *p* < 0.01.

## Data Availability

The data used to support the findings of this study are available from the corresponding author upon request.

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
