# Peer review of "Mucosa-Associated Lymphoid Tissue 1 Is an Oncogene Inducing Cell Proliferation, Invasion, and Tumor Growth via the Upregulation of NF-κB Activity in Human Prostate Carcinoma Cells"

_biomedicines, 2021, doi:10.3390/biomedicines9030250_

Round 1

Reviewer 1 Report

In this paper Tsui et al. report that expression of paracaspase MALT1 is upregulated in metastatic prostate cancer cell lines and in human prostate cancer tissues. The authors further show that overexpression of MALT1 in PC3 cells results in increased NF-kB-dependent secretion of IL-6 and CXCL5, as well as the invasion ability of PC3 cells or anchorage-independent growth of DU145 cells. There is also a very slight effect on proliferation of PC3 cells. Importantly, knock-down of MALT1 in PC3 cells inhibits tumor growth in an in vivo xenograft model. Finally, the authors also show that MALT1 expression is itself positively regulated by NF-kB, implicating a positive feedback mechanism. Although the idea is interesting, the experimental design suffers from several weaknesses and some strong statements are not sufficiently supported by the data (especially blots on NF-kB signaling).

Specific major comments:

  • Many of the data rely on ectopic overexpression of MALT1 in prostate cancer cell lines. I don’t think that further ectopic overexpression of MALT1 in prostate cancer cells, which according to the manuscript already overexpress MALT1, is a good approach to study the function of MALT1 in prostate cancer. How physiologically relevant is the ectopic overexpression still in such a case? Knockdown of MALT1 is therefore much more relevant. However, these experiments seem to suffer from a very poor knockdown efficiency (fig. 4A), preventing the authors to obtain conclusive results (e.g. effects on proliferation are very weak).
  • It is not clear to me how MALT1 overexpression would lead to MALT1 signaling (e.g. MALT1 overexpression in HEK293 cells to very high levels is not associated with any NF-kB signaling without co-expressing Bcl10 or one of the CARD-CC family members), as MALT1 signaling requires CBM complex formation which is known to be triggered by a mutagenic event in one of the CBM components or upstream signaling proteins, or upon an upstream extracellular stimulus. Thus, it would be most interesting to investigate whether MALT1 is indeed activated in these cells (e.g. by measuring MALT1 proteolytic activity). The authors should also discuss how overexpression could lead to MALT1 activation.
  • The authors mention that MALT1 is a paracaspase and even use a MALT1 inhibitor (MI-2) in some experiments. To link all the data, also the effect of a specific MALT1 inhibitor should be tested in the proliferation and invasion assays they show in Figure 2, especially as the effect of MALT1 overexpression on proliferation is very marginal.
  • The MI-2 inhibitor that is used in some experiments is unfortunately very unspecific to MALT1 (Bardet et al., 2018; PMID: 29359407) and also known to be very toxic. A more specific inhibitor, such as MLT-748, should therefore be used to be conclusive.
  • Figure 1: MALT1 is known to be constitutively expressed in all cell types. Why do the authors then not detect any signal in the non-metastatic cells? The identity of the MALT1 band should be confirmed by showing its disappearance upon MALT1 knockdown. Also, analysis of MALT1 expression in normal prostate cells would be of interest.
  • 3: The effect on NF-kB signaling is very weak and not convincing. E.g. there are no differences in P-IkBa for clone A2 if one looks at the figure in the supplementary file. Also the blot looking at Ikba degradation shows multiple bands, questioning the specificity of the signal. In fact, NF-kB activation is known to upregulated IkB levels, after the initial degradation.
  • It is known that MALT1 regulates IL-6 mRNA stability in an NF-kB-independent manner. Do the authors consider other NF-kB-independent activities of MALT1 in explaining the observed effects?
  • Figures 4B and 4C, why such a big difference in two different proliferation assays?
  • Figure 5A, the effect of MALT1 knockdown on pIkBa and IkBa levels, or p50/p65 nuclear translocation is not convincing, perhaps because MALT1 knockdown is inefficient. In fact, the only convincing knockdown of MALT1 is in Figure 4H (in vivo data), while in Figures 4A and 5A the knockdown of MALT1 is very inefficient. Why is the knockdown efficiency in the tumor xenografts suddenly much higher (100%) compared the efficiency in vitro (panel A)?
  • The in vitro data only show marginal effects on proliferation/invasion, while the in vivo data show much stronger effects. There is no correlation at all, questioning if the in vivo results really reflect the proposed role of MALT1 in vitro.
  • Why do the authors use NIK to activate NF-kB signaling? NIK is known to be part of the alternative NF-kB pathway while the authors show that MALT1 activates the classical NF-kB pathway. Also, it should be investigated if the MALT1 promotor does contain NF-kb binding sites. Conclusions on a positive feedback mechanism can therefore not be made from the shown experiments.
  • The quality of many antibodies that are used in western blotting or other techniques is known to be very poor. Therefore, specificity of the western blot results should be well documented. In this context, I assume that the extra file that was provided showing the raw data with different western blots is meant to document specificity. However, only small parts of the gels/blots are shown, which does not allow to evaluate specificity. Therefore, the authors should show the full blots covering a broad molecular weight range.

Minor comments:

  • Why do the authors decide to use transiently transfected PC3 cells for the reporter/proliferation/invasion experiments after showing expression of MALT1 in different prostate cancer cell lines? Also, why then switching to stably transfected DU145 cells?
  • Figure 2D please show actual ELISA values (pg/ml) instead of % secretion
  • Figure 3 C, I would prefer to see both Lamin B1 and GAPDH controls in both cytosolic and nuclear fraction similarly to Figure 5A, to exclude the possibility of incomplete fractionation.
  • Why do the authors use Matrigel assay for the invasion of PC3 cells and soft agar assay in the case of DU145?

Reviewer 2 Report

In this study the authors identify MALT1 to drive protumorigenic effects in an NF-kB-dependent manner in human prostate cancer cells and a mouse xenograft model. The topic of the manuscript is novel and interesting. In particular the identification of a putative positive feedback loop between MALT1 and NF-kB signaling, which might be a suitable target for prostate cancer therapy, is potentially important. Overall, the paper is well written and structured. However, the experimental data presented in this work appear rather premature and often lack several appropriate controls in order to fully support the authors’ conclusions.

Major points of concern:

  1. Figure 2: The authors convincingly show that ectopic overexpression of MALT1 leads to an upregulation of IL-6 and CXCL5 in PC-3 cells. Given that MALT1 is much higher expressed in LNCaP cells as compared to PC-3 cells (fig. 1), the authors should transiently silence MALT1 and analyze IL-6 and CXCL5 expression level. The authors should also demonstate the the MALT1-induced upregulation of IL-6 and CXCL5 is NF-kB mediated.

  1. Figure 3: Not clear why this is the only panel showing data from DU145 cells instead of PC-3 cells. Please explain.

  1. Figure 3C, D: The immunoblot analysis of NF-kB signaling components lacks proper control. Specifically, the authors must show that the cytosolic fraction is not contaminated by the nuclear fraction and vice versa (staining of cytosolic fraction for nuclear protein, staining of nuclear fraction for cytosolic protein). Also indicate how often the immunoblot analyses have been independently performed.

  1. Figure 4A: Knockdown of MALT1 is at least on protein level not convincing. Please indicate number of replicates and show a densitometric analysis, which also enables statistical evaluation.

  1. Figure 5A, lower panels, again appropriate controls for the purity of the protein fractions are missing. The authors should also indicate number of replicates and include a densitometric analysis in order to enhance the validity of these data

  1. Influence of MALT1 on NF-kB: Again, it would be interesting to see whether a knockdown of MALT1 affects NF-kB activity in high MALT1 expressing LNCaP cells. Immunoblots require appropriate controls and information about replicate numbers plus densitometry

Round 2

Reviewer 2 Report

The authors have adequately addressed my points of concern.